# Vision Model Pre-training on Interleaved Image-Text Data via Latent Compression Learning

**Chenyu Yang[1,3]\***, **Xizhou Zhu[1,2]\***, **Jinguo Zhu[3,6]\***, **Weijie Su[2,5]**, **Junjie Wang[3,7]**, **Xuan Dong[1,3]**, **Wenhai Wang[2,4]**, **Lewei Lu[3]**, **Bin Li[5]**, **Jie Zhou[1]**, **Yu Qiao[2]**, **Jifeng Dai[1,2]**✉

[1]Tsinghua University   [2]OpenGVLab, Shanghai AI Laboratory
[3]SenseTime Research   [4]The Chinese University of Hong Kong
[5]University of Science and Technology of China   [6]Xi'an Jiaotong University
[7]Beijing University of Posts and Telecommunications
`{yangcy23,x-dong21}@mails.tsinghua.edu.cn,`
`{zhuxizhou, jzhou, daijifeng}@tsinghua.edu.cn,`
`lechatelia@stu.xjtu.edu.cn, jackroos@mail.ustc.edu.cn, jjwang@bupt.edu.cn,`
`whwang@ie.cuhk.edu.hk, binli@ustc.edu.cn, qiaoyu@pjlab.org.cn, luotto@sensetime.com`

## Abstract

Recently, vision model pre-training has evolved from relying on manually annotated datasets to leveraging large-scale, web-crawled image-text data. Despite these advances, there is no pre-training method that effectively exploits the interleaved image-text data, which is very prevalent on the Internet. Inspired by the recent success of compression learning in natural language processing, we propose a novel vision model pre-training method called Latent Compression Learning (LCL) for interleaved image-text data. This method performs latent compression learning by maximizing the mutual information between the inputs and outputs of a causal attention model. The training objective can be decomposed into two basic tasks: 1) contrastive learning between visual representation and preceding context, and 2) generating subsequent text based on visual representation. Our experiments demonstrate that our method not only matches the performance of CLIP on paired pre-training datasets (e.g., LAION), but can also leverage interleaved pre-training data (e.g., MMC4) to learn robust visual representations from scratch, showcasing the potential of vision model pre-training with interleaved image-text data. Code is released at `https://github.com/OpenGVLab/LCL`.

## 1 Introduction

Over the past decade, ImageNet [34] pre-trained vision models have significantly advanced computer vision, continuously achieving breakthroughs in various vision tasks [7, 24, 10]. The success of ImageNet has inspired further exploration of better methods for pre-training vision models from scratch. Recently, the focus of pre-training has shifted from manually annotated data to large-scale, web-crawled image-text data. A key milestone in this shift is CLIP [55], which utilizes image-text pair data hundreds of times larger than ImageNet, delivering superior performance across various tasks and progressively becoming the mainstream method for vision model pre-training. Building on this trend, there is increasing interest in exploring interleaved image-text data, which is more prevalent on the Internet. Unlike the structured image-text pairs used in CLIP, this interleaved data is free-format and non-paired, larger in scale, and richer in textual information. Fully exploiting these interleaved image-text data is necessary for further improving vision model pre-training at scale.

---

\*Equal contribution. This work is done when Chenyu Yang, Jinguo Zhu, Junjie Wang and Xuan Dong are interns at SenseTime Research.✉Corresponding author: Jifeng Dai <daijifeng@tsinghua.edu.cn>.

38th Conference on Neural Information Processing Systems (NeurIPS 2024).

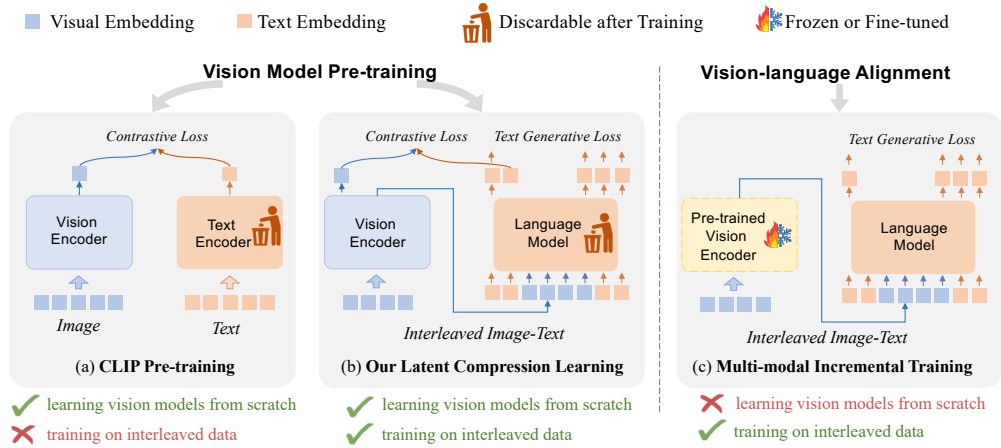

Figure 1: **Comparison of different training frameworks. (a)** Contrastive learning framework from CLIP [55] pre-trains vision encoders from scratch with image-text pairs, but it does not support interleaved data. **(b)** Our proposed LCL pre-training frameworkm can effectively pre-train vision encoders from scratch with interleaved image-text data. In these two frameworks, the text encoder or the language model that provides supervision can be optionally discarded during the transfer stage. **(c)** Multi-modal incremental training process uses interleaved image-text data to align the pre-trained vision encoder and the language model, but it cannot pre-train vision encoders from scratch.

Currently, no pre-training method can effectively utilize interleaved image-text data to pre-train vision models from scratch. In preliminary attempts [2, 29, 44] of using interleaved data, vision models were already pre-trained by CLIP on paired image-text data. Subsequent training on interleaved data primarily serves to align the pre-trained vision models with language models, thereby enhancing the multi-modal capabilities of the entire vision-language network. Therefore, it remains an important and open problem of how to effectively learn robust visual representation from scratch on interleaved image-text data.

A recent study [30] in Natural Language Processing (NLP) suggests that the success of modern language models originates from the compression of training datasets into model parameters. We believe that such compression learning is also applicable to the multi-modal field, except that the data to be compressed expands from structured plain texts to interleaved image-text data, where the images are of raw pixels and unstructured. Such compression learning should be revised to accommodate to the image data. Raw pixels are unstructured and often contain unnecessary and unpredictable details. Such details are irrelevant to the high-level semantic tasks and should be discarded in compression learning. Thus, we argue that compression learning on interleaved text-image data should be applied to latent image representation to better extract semantic abstracts.

In this paper, we propose a novel visual pre-training framework, named Latent Compression Learning. We first theoretically demonstrate that effective latent compression learning can be performed by maximizing the mutual information between outputs and inputs of a causal attention model. When applied to visual pre-training on interleaved image-text data, visual latents are extracted through a visual encoding network (such as ViT [20]), and then fed together with the text into a causal model. The optimization objective can be derived and decomposed into two parts: 1) *contrastive learning* between visual latent representation and their previous context to enhance semantic consistency, and 2) *auto-regressive prediction* to learn the predictability of visual representation for subsequent text. These two training objectives complement each other. For images, the learned latent representation retain information that can be predicted from previous contexts and information needed for predicting subsequent contexts, thus providing effective visual pre-training.

In the experiments, various interleaved and paired pre-training methods are evaluated. The evaluation is conducted through transfer learning on multiple tasks, including image classification, image-text retrieval, image captioning, and visual dialogue. The pre-training datasets include the widely used image-text paired LAION-400M [57] and the image-text interleaved MMC4 [88] and Obelics [36]. In addition, we also re-organize an interleaved version of LAION-Random and a paired version of MMC4-Pair, to facilitate the comparison between interleaved and paired pre-training methods under the same data source. Experiment results show that our LCL pre-training can achieve the same

performance as CLIP on paired pre-training data and can better utilize interleaved pre-training data. Our results also demonstrate the effectiveness of using interleaved image-text data to learn robust visual representation from scratch, and the potential of compression learning for visual pre-training.

## 2 Related Work

**Vision-centric Pre-training Methods.** Supervised Pre-training on large-scale annotated datasets [24, 62, 48, 82] has remained the mainstream method for a long time, and has been favored by various vision tasks [9, 24, 79, 7] demonstrating strong performance. Self-Supervised Pre-training has gained significant popularity due to its advantage of utilizing unlabeled data. BEiT [6] follows the methodology of BERT [19] by randomly masking image tokens and reconstructing them as targets. MAE [27] and SimMIM [80] directly use masked pixels as reconstruction targets, making the pre-training process more straightforward and efficient. Weakly-Supervised Pre-training leverages image-hashtag [50, 76, 63] and image-text datasets [71, 61, 8, 57], which rely on noisy text supervision from the internet. For image-hashtag datasets, related works [50, 63] have shown comparatively good performance across various transfer learning settings. In the case of image-text datasets, early efforts [3, 42, 49, 64, 66, 67, 70, 13, 39] focused on learning general visual-linguistic representation. Recently, exemplified by CLIP [55], methods [55, 32] have been developed that involve pre-training through aligned visual-linguistic representation, achieving outstanding results in image classification tasks. Other works like M3I Pre-training [65] propose a unified framework that integrates multiple pre-training strategies with data from various modalities and sources. Currently, weakly supervised pre-training from web-scale text supervision has become the core of multi-modal understanding, but existing methods have not yet to leverage the most widespread interleaved image-text data for training visual representation from scratch.

**Interleaved Image-Text Incremental Pre-training.** Training using Interleaved Image-Text Data (IITD) has recently garnered significant attention due to the vast amount of such data available online. Recent works[87, 68, 69, 23, 4] such as Flamingo [2] and KOSMOS-1 [29] perform incremental learning on non-public IITD based on previously pre-trained vision and language model parameters as initialization, training text generation models with multi-modal understanding capabilities. With the continuous advancement in the field and the proliferation of public IITD (*e.g.*, MMC4 [88] and OBELICS [36]), numerous models [83, 85, 14] capable of multi-modal understanding have emerged. However, these efforts only perform incremental pre-training on IITD and only analyze the usage of IITD on multi-modal dialogue models. Whether IITD contributes to learning robust visual representation from scratch remains unknown.

**Compression Learning in NLP.** The perspective [59, 60] that compression is closely connected to intelligence has a long history. A common compression method is arithmetic coding [56, 52], which is a practical approach that uses probability models for optimal data encoding. Some studies [28, 51, 38, 31, 37] argue that compression and intelligence are fundamentally equivalent. With the popularity of large language models, the equivalence of language modeling and compression has once again drawn widespread attention, prompting numerous explorations. Recently, [18] demonstrated through examples that language models serve as universal compressors. [30] posited that language modeling is equivalent to compressing a dataset into model parameters and proposed that compression efficiency is linearly related to model capabilities. Although compression learning has been proven effective in the field of NLP, it is not clear whether this approach can be extended to other fields.

**Multi-modal Large Models.** Text supervision pre-trained vision models [55, 32, 40] are widely utilized and have exhibited superior performance in tasks ranging from image retrieval and image classification to captioning. Currently, the most popular and closely watched application area is multi-modal dialogue. Multi-modal dialogue models [2, 41, 86, 17, 25, 21, 53] primarily rely on a powerful pre-trained vision encoder [55, 22] and text decoder [16, 74, 75]. The usage of pre-trained vision encoder can generally be divided into two categories: the majority [2, 72], represented by LLaVA [46], employ a strategy where the pre-trained vision encoder is frozen, and only the subsequent adapters and language models are trained. A minority, exemplified by Qwen-VL [5, 53], utilize high-quality image-text dialogue data to continue fine-tuning the vision model. These two training strategies correspond to the two evaluation methods used in this paper to assess the performance of vision models.

## 3 Method

### 3.1 Latent Compression Learning

**Auto-regressive Language Modeling as Compression Learning.** Recent works [18, 30] have shown that auto-regressive language modeling is equivalent to compression learning. Suppose $g_\phi$ is a language model (LM) with learnable parameters $\phi$. Given an input text sequence $x = (\texttt{}, x_1, x_2, \ldots, x_N)$, where $\texttt{}$ is a special token indicating the beginning of text, the model outputs $y = g_\phi(x) = (y_1, y_2, \ldots, y_N)$ predicting the next token based on preceding context, *i.e.*, $\hat{x}_k = y_k = g_\phi(x)_k$. The approximate probability of $x$ estimated by $g_\phi$ is $q(x) = \prod_{k=1}^{N} q(x_k | y_k = g_\phi(x)_k)$. The model is optimized with NLL-loss, which equals to minimizing the the cross-entropy between the data distribution $p$ and model distribution $q$:

$$H(p, q) = \mathbb{E}_{x \sim p} \left[ -\sum_{k=1}^{N} \log q(x_k | y_k = g_\phi(x)_k) \right]. \tag{1}$$

Notice that $H(p, q)$ is actually the optimal expected code length encoding $p$ by $q$, minimizing $H(p, q)$ just means compressing the data into the model parameters.

**Latent Compression for Interleaved Image-Text Data.** We believe that the compression principle could apply to multi-modal domain, specifically, to train vision-language models by compressing interleaved image-text data. However, instead of directly dealing with pixel values, we turn to compress high-level image representation for the following reasons: 1) high-level representation can extract useful information from raw pixels while discarding those unpredictable image details. 2) the learned visual representation will align with text semantics, making it possible to perform effective visual pre-training with interleaved image-text data.

Specifically, let $x = (\texttt{}, x_1, x_2, \ldots, x_N)$ be an interleaved image-text sequence. To simplify the expression without loss of generality, we assume that there is only one image in the sequence. Sub-sequence $x_{i:i+M}$ are $M + 1$ image patch tokens of the input image (e.g., non-overlapping patches in ViTs) and others are text tokens. $I = \{i, i+1, \ldots, i+M\}$ denotes the indices of the image patches, and $T = \{1, \ldots, i-1, i+M+1 \ldots, N\}$ denotes the indices of text tokens. As shown in Fig. 2, to construct the sequence of latent representation $z = (\texttt{}, z_1, z_2, \ldots, z_N)$, for image patches, we use a parametric vision encoder $f_\theta$ (e.g., ViTs) to map the data sequence $x_{i:i+M}$ into latent variable $z_{i:i+M}$. For text tokens, we directly use one-hot vectors corresponding to their vocabulary ids as the latent codes. Then, the latent representation $z$ are fed into a causal attention model $g_\phi$ for latent compression by minimizing

$$H(p, q) = -\int p(z) \log q(z) = \mathbb{E}_{x \sim p} \left[ -\sum_{k=1}^{N} \int p(z_k | x) \log q(z_k | y_k = g_\phi \circ f_\theta(x)_k) \right], \tag{2}$$

where the $k$-th element of the output $y_k = g_\phi \circ f_\theta(x)_k$ predicts the next input latent $z_k$, $f_\theta$ is identity for text tokens for simplicity of annotation. When the compression only applied on text tokens, it degenerates to auto-regressive language modeling on interleaved image-text data used by previous methods (e.g., Kosmos and Flamingo).

However, direct optimizing Eq. (2) for learning informative latent representation is non-trivial, since Eq. (2) suffers from a naturally trivial solution of visual representation collapse, i.e. the image latent representation $z_{i:i+M}$ may be learned to be data-independent. In fact, as showed in Sec. 4.2, we have observed such visual representation collapse, when training from scratch on MMC4 dataset with Eq. (2) applied to text tokens only (i.e., auto-regressive language modeling).

**Maximizing Mutual Information for Latent Compression Learning.** Optimizing directly for latent compression in Eq. (2) may cause the visual representation collapse. A natural constraint is to maximize the representation entropy to prevent collapse. We find that combining latent compression and maximum entropy constraint is exactly equivalent to maximizing the mutual information between the model inputs and outputs.

Prior work [65] have shown the relationship between cross-entropy and mutual information in other pre-training tasks. Here, we derive this relationship in the latent compression task: maximizing the mutual information between the output $y$ and the input latent $z$ of the causal attention model $g_\phi$ is

equivalent to compressing $z$ by minimizing $H(p, q)$ in Eq. (2) meanwhile maximizing the entropy of each element $z_k$ in $z$:

$$
\begin{aligned}
I(y; z) &= \mathbb{E}_{x \sim p} \left[ \sum_{k=1}^{N} \int p(y_k|x) p(z_k|x) \log \frac{p(z_k|y_k)}{p(z_k)} \right] \\
&= \max_{q} \mathbb{E}_{x \sim p} \left[ \sum_{k=1}^{N} \int p(z_k|x) \log q\left(z_k|y_k = g_\phi \circ f_\theta(x)_k\right) \right] - \sum_{k=1}^{N} \int p(z_k) \log p(z_k) \\
&= - \min_{q} H(p, q) + \sum_{k=1}^{N} H(z_k),
\end{aligned}
\tag{3}
$$

where we use $p(y_k, z_k|x) = p(y_k|x) p(z_k|x)$ in the first step since $z_k$ and $y_k$ can be independently computed given input $x$, and $p(z_k|y_k)$ is estimated by an approximate parameterized distribution $q(z_k|y_k)$. For the derivation of the formula, please refer to [65].

Therefore, using $I(y; z)$ as the optimization objective can achieve latent compression while avoiding representation collapse of $z$ via the maximum entropy constraint. The compression of $z$ imposes the model to extract useful information and discard unpredictable information of the image. Meanwhile, maximizing $I(y; z)$ requires that each $y_k$ could obtain enough information from previous latent $z_{<k}$ to predict $z_k$. Each $z_k$ should carry predictable information. These guarantee that the image representation encode rich semantic information aligned with text. We suppose that the above properties learned by the image representation are desired for vision-language pre-training, thus we use Eq. (3) as our pre-training objective. Parameters $\phi$ and $\theta$ are be jointly optimized under this objective. Intuitively, the vision encoder $f_\theta$ learns to represent images by high-level abstract, and the causal attention model $g_\phi$ learns to compress this high-level abstract of the dataset.

## 3.2 Training Loss

In this sub-section, we demonstrate how Eq. (3) is decomposed into training tasks and losses. Firstly, $I(y; z)$ can be decomposed as a cross-entropy term and an entropy term in the following two symmetric ways (see Appendix B for detailed derivation):

$$
I(y; z) = \sum_{k=1}^{N} - \min_{q_1} \mathbb{E}_{x \sim p} \left[ H\left(\delta\left[z_k = f_\theta(x)_k\right], q_1\left(z_k|y_k = g_\phi \circ f_\theta(x)_k\right)\right) \right] + H(z_k), \tag{4}
$$

$$
I(y; z) = \sum_{k=1}^{N} - \min_{q_2} \mathbb{E}_{x \sim p} \left[ H\left(\delta\left[y_k = g_\phi \circ f_\theta(x)_k\right], q_2\left(y_k|z_k = f_\theta(x)_k\right)\right) \right] + H(y_k). \tag{5}
$$

Since given the input $x$, latent $z_k$ and $y_k$ are independent and deterministic (*i.e.*, determined by $f_\theta$ and $g_\phi$), yielding $p(y_k, z_k|x) = \delta\left[z_k = f_\theta(x)_k\right] \cdot \delta\left[y_k = g_\phi \circ f_\theta(x)_k\right]$. $\delta[\cdot]$ is delta distribution. In Eq. (4), $p(z_k|y_k)$ is estimated by a parameterized distribution $q_1(z_k|y_k)$, which approximates the distribution of $z_k$ given the model's prediction $y_k$. Similarly, in Eq. (5), $p(y_k|z_k)$ is estimated by $q_2(y_k|z_k)$, the predicted distribution of $y_k$ given $z_k$. Therefore, maximizing mutual information can be decomposed as follows: 1) The causal attention model $g_\phi$ learns to predict the next latent $z_k$ from the output $y_k$. 2) The learnable latent representation $z_k$ learns to predict $y_k$, which is the representation of its previous context. 3) The maximum entropy regularization avoids the collapse of $z_k$ and $y_k$.

In the following, we show that the cross-entropy terms in $I(y; z)$ can be achieved by two common training tasks and loss functions, while the entropy constraints are implicitly satisfied.

**Contrastive Learning between Image Representation and Preceding Context.** For image latent $z_k$ and the corresponding $y_k$ representing the semantics of its preceding context, the objective defines a bidirectional prediction. We choose $q$ as Boltzmann distribution, *i.e.*, $q\left(z_k|y_k\right) \propto \exp(z_k^\top W_1^\top W_2 y_k / \tau)$ and $q\left(y_k|z_k\right) \propto \exp(y_k^\top W_2^\top W_1 z_k / \tau)$, where $\tau$ is the temperature, $W_1$ and $W_2$ are learnable linear projections. Consequently, the objective becomes the contrastive loss in two directions between $z_k$ and $y_k$, when setting $z_{k'}$ and $y_{k'}$ from other images as negative samples:

$$
L_{con} = - \sum_{k \in I} \log \frac{\exp(y_k^\top W_2^\top W_1 z_k / \tau)}{\sum_{k'} \exp(y_k^\top W_2^\top W_1 z_{k'} / \tau)} - \sum_{k \in I} \log \frac{\exp(z_k^\top W_1^\top W_2 y_k / \tau)}{\sum_{k'} \exp(z_k^\top W_1^\top W_2 y_{k'} / \tau)} \tag{6}
$$

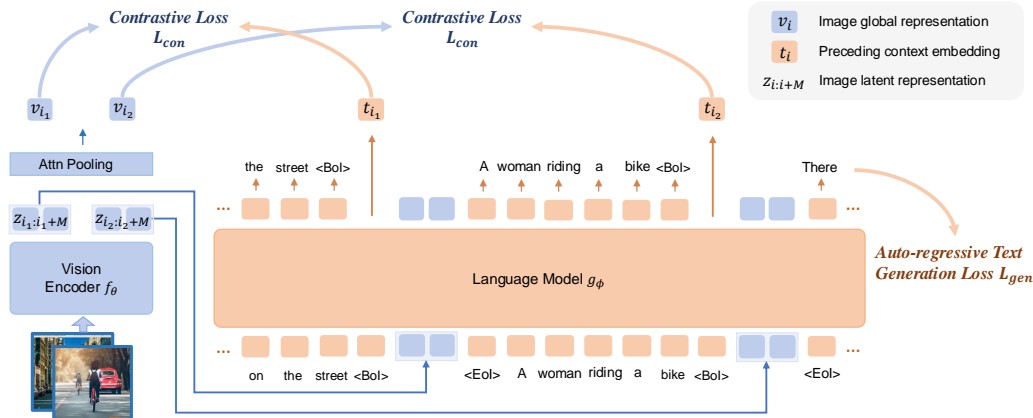

Figure 2: **Overview of our proposed Latent Compression Learning for vision model pre-training.** Image latent representation is extracted via a vision encoder and subsequently input into a language model alongside textual embedding. Two complementary losses are utilized to learn robust visual representation from scratch on interleaved image-text data: a contrastive loss ensures consistency between the visual latent representation and its preceding context, while an auto-regressive loss enhances the predictability of visual representation for subsequent text.

Meanwhile, the contrastive loss also prevents $z_k$ and $y_k$ from being trivial representation by pulling them away from negative samples, implicitly appending the entropy regularization.

**Auto-regressive Text Generation.** For a text token, its latent code $z_k$ is a one-hot vector and is not learnable, so the objective only imposes $y_k$ to predict $z_k$ as in Eq. (4). We choose $q(z_k|y_k)$ as softmax over the output logits on the text vocabulary, i.e, $q(z_k|y_k) = z_k^\top \mathrm{softmax}(V y_k)$, where $V$ is the projection head of the language model. The objective corresponding to the text tokens is simply the objective of standard next token prediction with cross-entropy loss:

$$L_{gen} = -\sum_{k \in T} \log z_k^\top \mathrm{softmax}(V y_k) \tag{7}$$

The total training loss is defined as $L = \lambda L_{con} + L_{gen}$, where $\lambda$ is balancing weight.

**Relation to Previous Pre-training Tasks.** In our proposed pre-training framework, the contrastive task and generation task align the image representation with preceding context and subsequent context, respectively. Hence, combining these two pre-training tasks can fully leverage the semantic information contained in interleaved image-text data to supervise the learning of image representation. For previous pre-training tasks, 1) *Contrastive Language-Image Pre-training (CLIP)* [55] has a similar objective of maximizing the mutual information between corresponding image and text [65], but it can only be applied to paired image-text data. 2) *Contrastive Captioner (CoCa)* [81] combines a captioning (text generation) loss with the CLIP loss, but it cannot be applied to interleaved data, either. CLIP loss requires image-text pairs, while the captioner can only be conditioned on a single image can rather than flexible interleaved image-text contents. 3) *Auto-regressive Text Generation* task only leverages the semantic information in subsequent context to supervise the learning of image representation, while the preceding context is missing, and representation collapse cannot be avoided. Moreover, interleaved image-text data usually has information redundancy, *i.e.*, the image and its corresponding text may contain similar information. So models may rely on information from the text rather than the image for prediction. This is particularly true when the vision encoder is trained from scratch, resulting that the image representation are never focused and optimized. Our experiments in Sec. 4.2 confirms this analyze.

### 3.3 Architecture

The overview of model architecture when adopting our LCL is shown in Fig. 2. The interleaved image-text input sequence may contain multiple images. We adopt a Vision Transformer (ViT) [20] as the vision encoder, which encodes each image into a sequence of visual embeddings as its latent representation. The visual embeddings of each image are inserted at corresponding positions in the

interleaved sequence, and we introduce special tokens <BoI> and <EoI> to indicate the beginning and the ending positions, respectively. The combined visual and text embeddings are fed into a causal language model. The text generation loss is the standard cross-entropy loss over the output logits of the language model defined in Eq. (7). In the contrastive learning task, calculating the loss for each image token is extremely computationally expensive. To alleviate this problem, we consider utilizing one global representation per image instead of all latent representation for contrastive learning. Specifically, the global representation $v_i$ of each image is extracted from its latent representation $z_{i:i+M}$ through an attention pooling, which is a multi-head attention layer with a learnable query token. The output of the causal transformer model at <BoI> token, just before the image, is processed by a LayerNorm layer and a linear projection into $t_i$. The contrastive loss in Eq. (6) is calculated between $v_i$ and $t_i$.

## 4 Experiment

### 4.1 Experiment Settings

**Pre-train Data.** The datasets utilized in our pre-training encompass the image-text pair dataset LAION-400M [57], as well as the image-text interleaved datasets MMC4 [88] and OBELICS [36]. We also re-organize two datasets for fair comparison with CLIP. 1) LAION-Random as a interleaved dataset. Images from LAION-400M are randomly placed before or after their paired caption to form image-text sequences. 2) MMC4-Pair as a paired dataset. For images in MMC4, we select matching text from the bipartite graph matching results derived from CLIP similarity to generate pseudo image-text paired data.

**Implementation Details.** We adopt the same image transform in OpenCLIP [15] for pre-training and set the image size as $224 \times 224$ for all experiments. We employ ViT [20] as our vision encoder. ViT-L/14 is used for the main results, and ViT-B/16 is used for ablation studies. The language model follows the same architecture as OPT-125M [84] but is randomly initialized. By default, the contrastive loss balancing weight is set at $\lambda = 0.1$. AdamW optimizer with $\beta_1 = 0.9$, $\beta_2 = 0.95$ and a weight decay of 0.1 are used. We employ a cosine learning rate schedule with linear warmup and set the peak learning rate at 5e-4 for the LAION data and 3e-4 for others. The model for the main results is trained for 200k iterations with 64k images per batch on average. In the ablation study, the models are trained for 250k iterations with 8k images per batch.

**Vision Encoder Evaluation.** CLIP pre-trained models typically uses zero-shot retrieval performance for evaluation. However such zero-shot inference on retrieval tasks [55] is not suitable for other generative pre-training methods. To address this discrepancy, we choose to evaluate pre-trained vision models by transfer learning on various downstream tasks under two configurations: "frozen transfer" and "full transfer". In the frozen transfer setting, only the parameters outside of the vision model are trained. In the full transfer setting, all parameters are trained.

The evaluation is conducted on image classification, image-text retrieval, image caption, and multi-modal dialogue. Please see Appendix A for more data, implementation, and evaluation details.

### 4.2 Comparison with Vision Pre-training Methods

In this section, we show the superiority of our LCL when using interleaved image-text data for vision pre-training. However, existing vision pre-training methods only support paired data, so we choose MMC4 dataset with a paired set (MMC4-Pair) available for those methods. In addition, to confirm that it is is not trivial to use interleaved data for vision pre-training, we also consider existing methods to use interleaved data. Those methods are originally proposed as MLLM training methods, but here we test them for vision pre-training. Besides, on paired image-text data, we also compare our LCL with existing vision pre-training methods and show they are comparable. See Appendix A.3 and A.2 for detailed pre-training and evaluation settings.

We aggregate all the pre-training methods divide them into the following tasks: (1) Image-text contrastive (Con.) [55, 15, 32], (2) Image-text contrastive + image captioning (Con. + Cap.) [81], (3) Image-text contrastive + image captioning + image-text matching (Con. + Cap. + Mat.) [40, 41, 14], (4) Auto-regressive text generation (Gen.) [2, 29, 44], (5) Auto-regressive text generation + image regression (Gen. + Reg.) [69, 68, 72], (6) Mask data modeling (Mask.) [78]. We select one

Table 1: **Frozen transfer evaluations of vision models pre-trained on the MMC4 dataset.** Vision models are pre-trained from scratch for all methods. "IN-1k" denotes image classification on ImageNet [33]. "ret." and "cap." denote image-text retrieval and image captioning, respectively. * The method names refer to implementing those methods with our experiment setting but not their trained checkpoints. For pre-trainig tasks, **Con.** image-text contrastive; **Cap.** image captioning; **Mat.** image-text matching; **Mask.** mask data modeling; **Gen.** auto-regressive text generatiton; **Reg.** image (feature) regression. The names in parentheses refer to the pre-training task but not their trained checkpoints. † Note that CoCa and BLIP2 need to pass each sample through the language model 2 and 3 times, respectively, to perform multi-task learning.

| * Pre-training method (task) | Pre-training data | IN-1k | COCO ret. | | Flickr30k ret. | | COCO cap. | | NoCaps cap. | |
|---|---|---|---|---|---|---|---|---|---|---|
| | | acc-1 | TR@1 | IR@1 | TR@1 | IR@1 | B@4 | C | B@4 | C |
| CLIP (Con.) | MMC4-Pair | 74.8 | 46.4 | 32.5 | **76.2** | 60.0 | 23.9 | 82.9 | 29.6 | 78.4 |
| †CoCa (Con. + Cap.) | MMC4-Pair | **75.4** | **48.6** | **34.3** | 76.5 | **61.9** | 23.7 | 84.8 | 30.0 | 80.5 |
| †BLIP2 (Con. + Cap. + Mat.) | MMC4-Pair | 74.5 | 46.5 | 31.3 | 74.9 | 57.8 | 23.7 | 82.9 | 29.4 | 78.1 |
| BEiT3 (Mask.) | MMC4 | 73.3 | 45.1 | 30.6 | 73.2 | 57.1 | 23.3 | 81.4 | 29.5 | 76.7 |
| Flamingo (Gen.) | MMC4 | 24.0 | 10.6 | 5.6 | 17.7 | 10.8 | 8.7 | 18.1 | 15.0 | 18.5 |
| Emu (Gen. + Reg.) | MMC4 | 5.7 | 2.3 | 1.4 | 4.8 | 2.7 | 0.3 | 4.4 | 0.6 | 4.6 |
| **LCL (Ours)** | MMC4 | **75.2** | **48.5** | **34.5** | 76.3 | 60.4 | **24.4** | **87.5** | **31.0** | **82.5** |

Table 2: **Frozen transfer evaluations of vision models pre-trained on LAION dataset.** Vision models are pre-trained from scratch for all methods. * The method names refer to implementing those methods with our experiment setting but not their trained checkpoints. † Note that CoCa and BLIP2 need to pass each sample through the language model 2 and 3 times, respectively, to perform multi-task learning.

| * Pre-training method (task) | Pre-training data | IN-1k | COCO ret. | | Flickr30k ret. | | COCO cap. | | NoCaps cap. | |
|---|---|---|---|---|---|---|---|---|---|---|
| | | acc-1 | TR@1 | IR@1 | TR@1 | IR@1 | B@4 | C | B@4 | C |
| CLIP (Con.) | LAION-400M | **75.0** | 47.2 | 34.2 | 76.5 | 59.8 | 24.1 | 84.1 | 30.0 | 78.8 |
| †CoCa (Con. + Cap.) | LAION-400M | 75.2 | **48.6** | **34.8** | **76.9** | **61.4** | **24.6** | **88.2** | 30.4 | 82.9 |
| †BLIP2 (Con. + Cap. + Mat.) | LAION-400M | 74.0 | 47.4 | 31.5 | 75.9 | 57.9 | 23.6 | 85.0 | 30.0 | 77.8 |
| **LCL (Ours)** | LAION-Random | 75.1 | 48.3 | 34.3 | 76.8 | 59.6 | **24.4** | 88.1 | **31.3** | **84.2** |

representative method from each task. For fair comparison, we implement these methods on the same model and training data with available open-source codes or our reproduction.

**Pre-training on Interleaved Image-Text Data.** We conduct experiments on the MMC4 dataset [88] to demonstrate the effectiveness of our LCL on interleaved image-text data. For pre-training methods that only support paired data, MMC4-Pair is used.

The results are shown in Tab. 1. Our LCL pre-training method significantly outperforms all other methods in the caption tasks, indicating that we can effectively utilize the rich text context information in MMC4 interleaved data. On the other hand, our method is on par with the best paired pre-training methods on classification and retrieval tasks. Since the paired pre-training methods are directly optimized for retrieval, our comparable performance shows that the visual features are learned to be highly distinguishable. It is worth mentioning that for more general interleaved data, where no paired versions exist, these paired pre-training methods cannot be applied.

In addition, we observed that the two methods using auto-regressive text generation do not achieve good performance and feature collapse occurs. However, their text prediction training loss is actually close to ours. This suggests that these methods tend to rely on redundant text information rather than image information for subsequent text prediction. As discussed in Sec. 3.1, our approach can avoid such collapse.

**Pre-training on Paired Image-Text Data.** We conduct experiments on the LAION-400M dataset to show that LCL pre-training also performs well on paired image-text data without specific modification. Tab 2 shows that our method is comparable to paired pre-training methods on various tasks, indicating that all information in paired data is fully exploited.

Table 3: Transfer evaluation results of pre-trained ViT/L-14 on classification, retrieval and captioning tasks.

| Model | Pre-training data | Pre-training epoch | IN-1k | COCO ret. | | Flickr30k ret. | | COCO cap. | | NoCaps cap. | |
|---|---|---|---|---|---|---|---|---|---|---|---|
| | | | acc-1 | TR@1 | IR@1 | TR@1 | IR@1 | B@4 | C | B@4 | C |
| *frozen transfer* | | | | | | | | | | | |
| OpenAI CLIP | WIT-400M | 32 | 83.7 | 61.7 | 48.2 | 89.0 | 75.8 | 32.1 | 116.0 | 35.5 | 108.9 |
| OpenCLIP | LAION-400M | 32 | **82.1** | 59.5 | **46.0** | 86.9 | 74.2 | 31.0 | 111.5 | 34.8 | 106.0 |
| **LCL (Ours)** | LAION-400M | 32 | **82.2** | 59.6 | **46.2** | 86.7 | 74.0 | 31.4 | 112.3 | **35.0** | 106.7 |
| **LCL (Ours)** | LAION-400M + MMC4 | 16 | **82.0** | 60.0 | **46.0** | 87.6 | 74.6 | 32.0 | **113.7** | 35.1 | 107.1 |
| *full transfer* | | | | | | | | | | | |
| OpenAI CLIP | WIT-400M | 32 | 87.4 | 62.1 | 49.6 | 90.3 | 77.9 | 39.5 | 132.7 | 41.4 | 116.9 |
| OpenCLIP | LAION-400M | 32 | 86.2 | 61.7 | **47.5** | 87.7 | 76.3 | 38.6 | 128.9 | 39.9 | 112.7 |
| **LCL (Ours)** | LAION-400M | 32 | 86.1 | 61.9 | **47.5** | 87.6 | 76.1 | 39.1 | 129.7 | 40.2 | 113.5 |
| **LCL (Ours)** | LAION-400M + MMC4 | 16 | 86.1 | 62.2 | **47.6** | 88.1 | 76.2 | 39.5 | 130.9 | 40.6 | 113.8 |

Table 4: Transfer evaluation results of pre-trained ViT/L-14 on multi-modal benchmarks. The transfer adopts the downstream model and training pipeline of LLaVA-1.5 [46].

| Model | Pre-training data | Pre-training epoch | VQAv2 | GQA | VisWiz | SQA | POPE | MME | MMB | SEED$_I$ |
|---|---|---|---|---|---|---|---|---|---|---|
| *frozen transfer* | | | | | | | | | | |
| OpenAI CLIP | WIT-400M | 32 | 77.1 | 61.7 | 44.4 | 71.1 | 84.6 | 1486.9 | 65.1 | 64.6 |
| OpenCLIP | LAION-400M | 32 | 68.7 | 57.0 | 39.5 | 69.0 | **81.8** | 1266.2 | 54.2 | 55.8 |
| **LCL (Ours)** | LAION-400M + MMC4 | 16 | **70.7** | **57.4** | **41.4** | **69.7** | 81.7 | **1291.7** | **55.3** | **56.5** |
| *full transfer* | | | | | | | | | | |
| OpenAI CLIP | WIT-400M | 32 | 79.0 | 62.8 | 46.8 | 71.8 | 85.7 | 1576.8 | 68.9 | 67.9 |
| OpenCLIP | LAION-400M | 32 | 71.5 | **58.6** | 42.2 | 70.4 | **82.5** | 1345.4 | 58.5 | 59.5 |
| **LCL (Ours)** | LAION-400M + MMC4 | 16 | **73.4** | **58.8** | **44.2** | 71.0 | **82.5** | **1382.3** | **59.5** | **60.3** |

## 4.3 Comparison with Pre-trained Checkpoints

To further confirm the effectiveness of our proposed Latent Compression Learning (LCL), we compare our pre-trained model with existing checkpoints of pre-trained vision encoders. We use LCL to pre-train a ViT-L/14 with mixed data from the LAION-400M and MMC4. We compare it to the ViT-L/14 pre-trained by OpenCLIP [15] using the public LAION-400M dataset, and the ViT-L/14 pre-trained by OpenAI CLIP [55] with private data is listed as reference. The total number of images seen during pre-training is 13B for all models. We evaluate the pre-trained vision encoders by transferring to downstream tasks, *i.e.*, integrate the vision encoders into downstream task models and compare the fine-tuning results. More training details are in Appendix A.1 and evaluation details are in Appendix A.2.

Tab. 3 and Tab. 4 show the results of transfer evaluations. When both use LAION-400M as pre-training data, as with the previous experimental conclusions, LCL has similar performance to CLIP. When combined with MMC4, our method achieves better performance, especially on caption and multi-modal dialogue tasks.

There exist approaches achieving better results on some benchmarks. However, they either use larger vision encoders, more training data, or private data. We list those results as reference in Appendix C.

## 4.4 Ablation Study

**Latent Compression Learning on Different Datasets.** We apply LCL pre-training to more datasets to confirm its generalizability. As shown in Tab. 5, our method also achieves reasonable performance on interleaved dataset OBELICS. It is worth noting that the models trained on MMC4 and OBELICS have achieved similar performance to that on LAION, indicating that it is completely feasible to pre-train visual models only from interleaved data. Furthermore, using both LAION and MMC4 data during pre-training improves performance, suggesting that further improvements can be obtained by incorporating more image-text data. In this case, supporting interleaved data is a key advantage of our approach, enabling the use of more diverse image-text data for pre-training.

Table 5: **Frozen transfer evaluations of LCL pre-training on different datasets.** We ensured that all entries had seen the same number of images during pre-training to ensure fairness.

| Pre-training method | Pre-training data | IN-1k | COCO ret. | | Flickr30k ret. | | COCO cap. | | NoCaps cap. | |
|---|---|---|---|---|---|---|---|---|---|---|
| | | acc-1 | TR@1 | IR@1 | TR@1 | IR@1 | B@4 | C | B@4 | C |
| LCL (Ours) | LAION | 75.1 | 48.3 | 34.3 | 76.8 | 59.6 | 24.4 | 88.1 | 31.3 | 84.2 |
| | MMC4 | 75.2 | 48.5 | 34.5 | 76.3 | 60.4 | 24.4 | 87.5 | 31.0 | 82.5 |
| | Obelics | 73.9 | 47.0 | 33.2 | 75.1 | 58.7 | 23.3 | 84.8 | 29.9 | 77.6 |
| | LAION + MMC4 | **75.8** | **49.4** | **35.3** | **77.7** | **61.1** | **25.1** | **88.9** | **31.4** | **84.9** |

Table 6: **Ablations of the training loss and loss balancing weight in LCL .** Models are evaluated under frozen transfer setting.

(a) Training loss ablation.

| Training loss | COCO ret. | | COCO cap. | |
|---|---|---|---|---|
| | TR@1 | IR@1 | B@4 | CIDEr |
| Con. only | 46.4 | 32.7 | 23.8 | 82.7 |
| Gen. only | 10.3 | 5.4 | 8.5 | 17.8 |
| **LCL** | **48.5** | **34.5** | **24.4** | **87.5** |

(b) Loss balancing weight ablation.

| Con. weight $\lambda$ | COCO Ret. | | COCO Cap. | |
|---|---|---|---|---|
| | TR@1 | IR@1 | B@4 | CIDEr |
| 0.05 | 48.3 | 33.7 | 24.3 | 87.0 |
| **0.1** | **48.5** | **34.5** | **24.4** | **87.5** |
| 0.2 | 47.6 | 33.1 | 24.2 | 86.3 |
| 0.5 | 37.1 | 23.2 | 20.4 | 66.7 |

**Loss Balance in Latent Compression Learning.** Table 6a ablates the contrastive loss and generation loss used in LCL . Consistent with the previous analyses, LCL can achieve the best performance. Table 6b studies the appropriate loss balancing weights (multiplied by the contrastive loss). It turns out that $\lambda = 0.1$ will produce the best results. The performance drops significantly for larger $\lambda$ values, indicating that the optimization directions of the two losses are not completely consistent.

## 5 Conclusion

No existing work has explored vision model pre-training with interleaved image-text data. To this end, we propose Latent Compression Learning (LCL) framework that compresses interleaved image-text latent for vision pre-training. We theoretically show that latent compression is equivalent to maximizing the mutual information between the input and output of a causal model and further decompose this objective into two basic training tasks. Experiments demonstrate that our method is comparable to CLIP on paired pre-training datasets, and it effectively learns robust visual representations utilizing interleaved image-text data. Our work showcases the effectiveness of using interleaved image-text data to learn robust visual representation from scratch, and confirms the potential of compression learning for visual pre-training.

**Limitations**. Our experiments are constrained to a limited size of dataset and vision encoder, and the scaling property of our proposed method remains unexplored.

## Acknowledgments.

This work is supported by the National Key R&D Program of China (NO. 2022ZD0161300), by the National Natural Science Foundation of China (62376134).

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

# A Experimental Details

## A.1 Pre-training

**Data.** For the data in MMC4, we select images with a CLIP similarity to the matching text of 0.24 or higher. From documents containing at least one such image, we randomly choose up to 6 images to form an interleaved image-text sequence, utilizing all text from that document. If the sequence length exceeds 2048 tokens, the surplus is truncated while ensuring the integrity of both images and individual text segments, and then padded to the designated length. For OBELICS, we similarly restrict the number of images per document to between 1 and 6. We then sequentially extract 2048 tokens from the concatenated documents. If image tokens are truncated, the entire image is moved to the next sample sequence.

To construct interleaved image-text samples from the MMC4 dataset, we randomly place images either before or after their corresponding sentences, adhering to a 50% probability, thus generating a document-wise interleaved sequence of images and text. For the OBELICS corpus, individual documents are concatenated, and a sliding window strategy is employed to select each image-text sequence, maintaining a total length of 2048 tokens.

**Hyper-parameters.** Our pre-training configuration is shown in Tab. 7. The AdamW optimizer was employed for model training with the learning rate set to 3e-4 and the weight decay set to 0.1. Mixed numerical precision training with bfloat16 is also employed to stabilize the optimization process. Furthermore, we set a drop-path [35] rate linearly increasing to 0.2, and use layer-scale [73] for stable training.

Table 7: **Hyper-parameters in pre-training.**

| | |
|---|---|
| optimizer | AdamW |
| learning rate | 3e-4 |
| weight decay | 0.1 |
| optimizer momentum | $\beta_1, \beta_2 = 0.9, 0.95$ |
| lr schedule | cosine decay |
| warmup | 2k |
| numerical precision | bfloat16 |
| train steps | 20k |
| batch size (in images) | 64k |
| drop path | 0.2 |

## A.2 Evaluation

**Transfer Tasks.** We conduct our performance evaluation of pre-trained on image classification, image-text retrieval, text generation tasks with multimodal inputs (*i.e.*, image captioning and multimodal dialogue). Their model architecture in transfer learning are illustrated in Fig. 3.

For closed-set image classification, a lightweight classifier with a randomly initialized attention pooling layer, followed by a layer normalization layer and a linear layer, is appended to the top of the pre-trained vision model. In the "frozen transfer" scenario, only the parameters of the added classifier are trainable, similar to the linear probing strategy. Conversely, in the "full transfer" approach, all parameters, including those of the pre-trained vision encoder, are adjustable.

Regarding the image-text retrieval task, we discard the pre-trained text encoder and apply contrastive learning to the pretrained vision encoder with a newly introduced text encoder. Images are processed through the vision encoder and a randomly initialized attention pooling layer to generate a global image embedding. Textual captions are processed through the text encoder, utilizing the feature of the final token as the text embedding representing the input caption. An additional linear layer facilitates the dimensional alignment between the image and text embeddings, enabling their use in contrastive learning. During "frozen transfer", the attention pooling layer on the vision transformer and the text encoder are trainable; similarly, in "full transfer", all parameters, including the vision encoder, can be optimized.

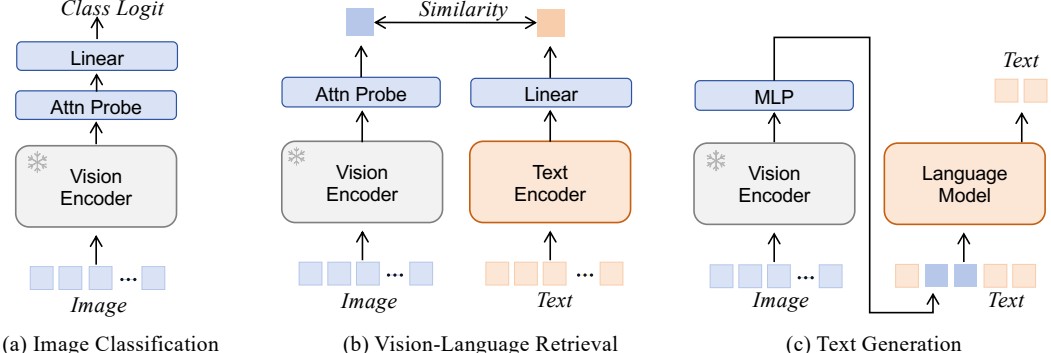

| (a) Image Classification | (b) Vision-Language Retrieval | (c) Text Generation |

Figure 3: **Illustration of "frozen transfer" evaluation**. The vision encoder is frozen during transfer tuning. (a) Image classification: an attention probe and a linear classifier are built upon the vision encoder. (b) Image-text retrieval: an attention probe is used to extract global visual feature, which is trained to align with the text feature from the text encoder. (c) Text generation: an MLP is utilized to align the visual feature with the text embedding space, and the multi-modal embedding is fed into the language model for auto-regressive text generation.

For text generation tasks with multimodal inputs, we adopt image captioning and multi-modal dialogue benchmarks and employ prevalent architectures like those in [46]. Specifically, a pretrained LLM for text generation is integrated on top of the pre-trained vision model, incorporating an MLP layer to adjust the dimensions of the visual embeddings. During the "frozen transfer" evaluation, the parameters of the vision model remain fixed, while in the 'unfreezing' phase, these parameters are permitted to undergo training.

**Implementation Details.** Vision encoder evaluation with "frozen transfer" and "full transfer" configurations includes fine-tuning training and benchmark evaluation. Implementation details of each transfer task are list below, and the hyper-parameters involved are listed in Tab. 8.

- *Image classification.* Model is trained on the ImageNet-1K [33] *train* split and evaluated on *val* split. We follow the attention probe setting introduced by [12] for "frozen transfer", and the full fine-tune setting in [43] for "full transfer".

- *Image-text retrieval.* Model is trained on a combination dataset comprised of CC12M [61], CC3M [61], and SBU [77], and is tested on the MSCOCO [11] *karpathy-test* split and Flickr30k [54] *test* split. The model is trained with Adamw optimizer for 5000 iterations. The learning rate is set at 1e-3 and 1e-5 for parameters without initialization and with initialization, respectively.

- *Image captioning.* Model is trained on a subset of the LAION-COCO [58] dataset, which includes 10 million samples, and evaluation is performed on the MSCOCO [11] *karpathy-test* split and NoCaps [1] *val* split. Here, the model is trained for 20,000 iterations with a learning rate of 1e-4. Additionally, a droppath technique is employed with a ratio of 0.2 in the vision model to mitigate overfitting.

- *Multi-modal dialogue.* We follow a two-stage training process similar to that used in LLAVA-1.5 [45]. Initially, paired data with 558K samples is used to train an MLP projector to align the Vision Transformer (ViT) with the pretrained LLM. Subsequently, the model undergoes instruction tuning on multimodal dialogue datasets with 665k samples. Both the alignment training and instruction tuning phases are conducted over a single epoch, with learning rates set at 1e-3 and 2e-5, respectively. Evaluations are then performed on multimodal dialogue benchmark, *e.g.*, MMBench [47] and VQAv2 [26].

## A.3  Ablation Experiments

The effectiveness of our LCL are validated by conducting ablation experiments mainly on two corpora: LAION and MMC4. The experimental hyper-parameters involved are shown in Tab. 9. We found that the optimal learning rate for the LAION dataset is 5e-4, while for the MMC4 dataset, a slightly lower

Table 8: **Hyper-parameters in transfer evaluation.**

| Hyper-parameter | Classification | Retrieval | Image Captioning | Multi-modal dialogue | |
| --- | --- | --- | --- | --- | --- |
| | | | | stage1 | stage2 |
| train dataset | IN-1K train | CC12M, CC3M, SBU | LAION-COCO | LCS-558k | LLaVA-SFT |
| test dataset | IN-1K val | COCO,Flickr30k | COCO,Nocaps | MMbench, VQAv2, GQA... | |
| optimizer | | | AdamW | | |
| learning rate | 1e-4 | 1e-3 | 1e-4 | 1e-3 | 2e-5 |
| weight decay | | | 1e-4 | | |
| optimizer momentum | | | $\beta_1, \beta_2 = 0.9, 0.95$ | | |
| learning rate schedule | | | cosine decay | | |
| warmup | 1500 | 500 | 1000 | 30 | 156 |
| train steps | 14k (90ep) | 5k | 20k | 1091 | 5198 |
| batch size (in images) | 8192 | 16k | 512 | 512 | 128 |

rate of 3e-4 proves most effective. We speculate that this is because MMC4 corpus contains relatively higher noise. Most of the original settings in the large-scale pre-training are retained in the ablation, with the exception of reducing the batch size by a factor of 8 to decrease the computational overhead.

Table 9: **Hyper-parameters in ablation study.**

| Hyper-parameter | LAION | MMC4 |
| --- | --- | --- |
| optimizer | AdamW | |
| learning rate | 5e-4 | 3e-4 |
| weight decay | 0.1 | |
| optimizer momentum | $\beta_1, \beta_2 = 0.9, 0.95$ | |
| learning rate schedule | cosine decay | |
| warmup | 2k steps linear | |
| numerical precision | bfloat16 | |
| train steps | 25k | |
| batch size (in images) | 8k | |
| drop path | 0.2 | |

### A.4 Experiments Compute Resources

Pre-training used 512 A800 GPUs and took 5 days.

## B Theoretical derivation details

Using the notation defined in Sec. 3, the mutual information of the output of the language model ($y$) and the latent representation ($z$) can be described as follows:

$$I(y; z) = \sum_{k=1}^{N} I(y_k; z_k)$$

$$= \sum_{k=1}^{N} \int p(y_k, z_k) \log \frac{p(y_k, z_k)}{p(y_k)p(z_k)} \, dy_k \, dz_k \tag{8}$$

$$= \sum_{k=1}^{N} \mathbb{E}_{x \sim p} \left[ \int p(y_k \mid x)p(z_k \mid x) \log \frac{p(y_k, z_k)}{p(y_k)p(z_k)} \, dy_k \, dz_k \right] \tag{9}$$

We can derive Eq. 9 from Eq. 8 because once the interleaved image-text input sequence $x$ is given, the output $y_k$ and the latent representation $z_k$ can be computed independently:

$$p(y_k, z_k \mid x) = p(y_k \mid x)p(z_k \mid x)$$

From Eq. 9, we can further decompose the mutual information into a cross-entropy component and an entropy component in two symmetric ways. One approach involves the output token $y_k$ predicting the next latent representation $z_k$, along with the entropy of $z_k$:

$$I(y; z) = \sum_{k=1}^{N} \mathbb{E}_{x \sim p} \left[ \int p(y_k \mid x) p(z_k \mid x) \log \frac{p(z_k \mid y_k)}{p(z_k)} \, dy_k \, dz_k \right] \tag{10}$$

$$= \sum_{k} \mathbb{E}_{x \sim p} \left[ \delta(z_k = f_\theta(x)_k) \log P(z_k \mid y_k = g_\phi \circ f_\theta(x)_k) \right] - \int p(z_k) \log p(z_k) \tag{11}$$

The other approach considers how the latent representation $z_k$ approximates the previous context $y_k$ and includes the entropy of the output $y_k$:

$$I(y; z) = \sum_{k=1}^{N} \mathbb{E}_{x \sim p} \left[ \int p(y_k \mid x) p(z_k \mid x) \log \frac{p(y_k \mid z_k)}{p(y_k)} \, dy_k \, dz_k \right] \tag{12}$$

$$= \sum_{k=1}^{N} \mathbb{E}_{x \sim p} \left[ \delta(y_k = g_\phi \circ f_\theta(x)_k) \log P(y_k \mid z_k = f_\theta(x)_k) \right] - \int p(y_k) \log p(y_k) \tag{13}$$

Note that the reason for transitioning from Eq. 10 to Eq. 11 and from Eq. 12 to Eq. 13 is because, given the input sequence $x$, the outputs $y_k$ and $z_k$ are determined as follows:

$$p(y_k \mid x) = \delta \left[ y_k = g_\phi \circ f_\theta(x)_k \right]$$
$$p(z_k \mid x) = \delta \left[ z_k = f_\theta(x)_k \right]$$

## C   Supplementary Benchmark Results

We present supplementary results on classification, retrieval and captioning tasks (Tab. 10) and multi-modal benchmarks (Tab. 11).

## D   Broader Impacts

This work may share the common negative impacts of large-scale vision training. The data used in pre-training may contain dataset bias, and raise ethical concerns. It may also require large computational resources, which consume lots of electricity and result in increased carbon emissions.

Table 10: Supplementary transfer evaluation results on classification, retrieval and captioning tasks. * Reproducing or using open-source code. "Repro. CoCa": reproduced CoCa. Results with larger vision encoders, more training data, or private data are grayed out as reference.

| Vision pre-training method | Support interleave data | Vision encoder | Pre-training data | Seen samples | Data public | IN-1k frozen | IN-1k finetune | COCO ret. frozen TR@1 | COCO ret. frozen IR@1 | COCO cap. finetune CIDEr |
|---|---|---|---|---|---|---|---|---|---|---|
| *Comparison with existing vision pre-training methods on paired data* | | | | | | | | | | |
| LCL (ours) | ✓ | ViT-B (86M) | LAION-400M | 2B | ✓ | 75.0 | - | 48.3 | 34.3 | - |
| *OpenCLIP | | ViT-B (86M) | LAION-400M | 2B | ✓ | 75.0 | - | 47.2 | 34.2 | - |
| *Repro. CoCa | | ViT-B (86M) | LAION-400M | 2B | ✓ | 75.2 | - | 48.6 | 34.8 | - |
| *Comparison with MLLM training methods (used for vision pre-training) on interleaved data* | | | | | | | | | | |
| LCL (ours) | ✓ | ViT-B (86M) | MMC4 | 2B | ✓ | 75.2 | - | 48.5 | 34.5 | - |
| *BEiT3 | ✓ | ViT-B (86M) | MMC4 | 2B | ✓ | 73.3 | - | 45.1 | 30.6 | - |
| *OpenFlamingo | ✓ | ViT-B (86M) | MMC4 | 2B | ✓ | 24.0 | - | 10.6 | 5.6 | - |
| *Emu | ✓ | ViT-B (86M) | MMC4 | 2B | ✓ | 5.7 | - | 2.3 | 1.4 | - |
| *Comparison with existing pre-trained checkpoints or reported results* | | | | | | | | | | |
| LCL (ours) | ✓ | ViT-L (304M) | LAION-400M | 13B | ✓ | 82.2 | 86.1 | 59.6 | 46.2 | 129.7 |
| LCL (ours) | ✓ | ViT-L (304M) | LAION-400M + MMC4 | 13B | ✓ | 82.0 | 86.1 | 60.0 | 46.0 | 130.9 |
| OpenCLIP | | ViT-L (304M) | LAION-400M | 13B | ✓ | 82.1 | 86.2 | 59.5 | 46.0 | 128.9 |
| OpenAI CLIP | | ViT-L (304M) | WIT-400M | 13B | | 83.7 | 87.4 | 61.7 | 48.2 | 132.7 |
| CoCa | | ViT-G (1B) | JFT-3B + ALIGN | 33B | | 90.6 | 91.0 | - | - | 143.6 |

Table 11: Supplementary results of multi-modal benchmarks. † Training data of the teacher model is included. Approaches are grayed out as reference if they use stronger vision encoders or LLM, more training data, or private data.

| Vision pre-training method | Vision encoder | Pre-training samples | Pre-training data public | MLLM method | LLM size | MLLM training samples | MLLM data public | VQAv2 | GQA | VizWiz | SQA | POPE | MME | MMB | SEED$_I$ |
|---|---|---|---|---|---|---|---|---|---|---|---|---|---|---|---|
| *Our transfer evaluation* | | | | | | | | | | | | | | | |
| **LCL (ours)** | ViT-L (304M) | 13B | ✓ | LLaVA-1.5 | 7B | 1.2M | ✓ | 70.7 | 57.4 | 41.4 | 69.7 | 81.7 | 1291.7 | 55.3 | 56.5 |
| OpenCLIP | ViT-L (304M) | 13B | ✓ | LLaVA-1.5 | 7B | 1.2M | ✓ | 68.7 | 57.0 | 39.5 | 69.0 | 81.8 | 1266.2 | 54.2 | 55.8 |
| OpenAI CLIP | ViT-L (304M) | 13B | | LLaVA-1.5 | 7B | 1.2M | ✓ | 77.1 | 61.7 | 44.4 | 71.1 | 84.6 | 1486.9 | 65.1 | 64.6 |
| *Advanced MLLMs as reference* | | | | | | | | | | | | | | | |
| OpenAI CLIP | ViT-L-336 (304M) | 13B | | LLaVA-1.5 | 7B | 1.2M | ✓ | 78.5 | 62.0 | 50.0 | 66.8 | 85.9 | 1510.7 | 64.3 | 65.4 |
| OpenAI CLIP | ViT-L-336 (304M) | 13B | | LLaVA-NEXT | 7B | 1.3M | | 81.8 | 64.2 | 57.6 | 70.1 | 86.5 | 1519 | 67.4 | 70.2 |
| OpenAI CLIP | ViT-L (304M) | 13B | | InternLM-XC2 | 7B | ~20M | | - | - | - | - | - | 1712 | 79.6 | 75.9 |
| OpenCLIP | ViT-L (304M) | 34B | | Qwen-VL-Chat | 7B | ~1.5B | | 78.2 | 57.5 | 38.9 | - | - | 1487.5 | - | - |
| EVA-CLIP | ViT-G (1B) | ~28B† | | Emu-I | 13B | ~83M | | 62.0 | 46.0 | 38.3 | - | - | - | - | - |
| EVA-CLIP | ViT-E (4.4B) | ~28B† | | Emu2-Chat | 33B | ~160M | | 84.9 | 65.1 | 54.9 | - | - | - | - | - |
| EVA-CLIP | ViT-E (4.4B) | ~28B† | | CogVLM-Chat | 7B | ~1.5B | | 82.3 | - | - | 91.2 | 87.9 | - | 77.6 | - |
| InternViT | ViT-6B | ~29B | | InternVL | 7B | ~1.0B | | 79.3 | 62.9 | 52.5 | - | 86.4 | 1525.1 | - | - |
| InternViT | ViT-6B | ~29B | | InternVL 1.5 | 20B | ~25M | | - | 65.7 | 63.5 | 94.0 | 88.3 | 1637 | 82.2 | 76.0 |

