# OpenReview forum: "Vision Model Pre-training on Interleaved Image-Text Data via Latent Compression Learning"
_NeurIPS.cc/2024/Conference — NeurIPS 2024 poster_

### Official Review · Reviewer_vQis · 2024-07-08

**Soundness:** 3
**Presentation:** 3
**Contribution:** 3
**Rating:** 6
**Confidence:** 4

**Summary:**

This paper introduces a vision backbone pre-training method named Latent Compression Learning (LCL) to utilize interleaved image-text data. The proposed LCL approach maximizes mutual information between the inputs and outputs of a GPT-like model in autoregressive manner. The proposed method integrate both discriminative and generative objectives by contrasting preceding context and generate subsequent text based on visual representation. The extensive experiments demonstrate that LCL not only matches the performance of existing models like CLIP on paired datasets (e.g., LAION) but also effectively leverages interleaved pre-training data (e.g., MMC4) to learn robust visual representations from scratch.

**Strengths:**

1. The paper is well written and easy to follow.

2. The paper introduces a new pre-training method, Latent Compression Learning (LCL), which utilizes interleaved image-text data for visual backbone pre-training for the first time. And this can effectively leveraging large-scale web-crawled data, which is easier to crawl compared to the image-text pairs.

3. Extensive experiments are conducted, demonstrating the effectiveness of the proposed method on both paired datasets (e.g., LAION) and interleaved datasets (e.g., MMC4).

**Weaknesses:**

1. From Table 5, it appears that solely leveraging image-text pairs with LCL does not provide benefits over the CLIP baseline. However, when using the MMC4 dataset, which is manually composed of interleaved text, there is significant performance improvement on downstream tasks. I am curious whether this performance gain results from the increased number of training samples (i.e., the total number of images used during training).

2. According to Table 3, utilizing original interleaved datasets such as Obelics does not yield any performance gain. In comparison, the MMC4 dataset requires more computation for data filtering with the CLIP score and the use of image-text pairs to create interleaved data. It is unclear how to efficiently utilize the original interleaved data directly crawled from the web. Do you have any insights on the differences between these two types of interleaved datasets?

**Questions:**

1. The scaling behavior is not demonstrated. While the author has shown the effectiveness of training on MMC4 and Laion-400M, it remains unclear how the model performs and correlates across different dataset scales. Understanding this could provide valuable insights into the feasibility and performance of scaling the proposed method to larger datasets, such as DataComp 12.8B and Laion 2B.

2. Can you show the seen samples of each model? It would be helpful for readers understanding the scale of model's training.

**Limitations:**

The authors have addressed the limitation in their manuscript.

---

> ### Author Rebuttal · Authors · 2024-08-07
>
> Thanks for your good questions and constructive suggestions.
> ___
> **Q1:** In Tab. 5, LCL is on par with CLIP baseline solely with image-text pairs but is significantly better when using the MMC4 dataset. Whether this performance gain is from the increased number of training samples.
>
> **A1:** It is reasonable that our LCL is comparable to CLIP on image-text pairs. Our LCL uses the simply constructed LAION-Random dataset (see Section 4.1), degrading to using half of the data for contrastive learning and the other half for generation. Compared to CLIP which uses all data for contrastive learning, there is no significant advantage or disadvantage.
>
> When incorporating MMC4 data, the significant improvement comes from increased data diversity. As mentioned in Section 4.3, all models in Tab. 5 are exposed to the same total number of training images. The number of images per epoch doubles when using LAION-400M + MMC4. Tab. 3 also shows that using MMC4 and LAION-400M alone are on par. This suggests that data diversity is the primary factor.
>
> Therefore, the advantage of our LCL is its ability to use larger-scale and more diverse interleaved image-text data, while not suffering performance losses when using paired data.
> ___
> **Q2:** In Tab. 3, using original interleaved datasets such as Obelics does not yield performance gain. MMC4 uses CLIP score to filter image-text pairs and create interleaved data. Any insights on the differences between these two types of interleaved datasets?
>
> **A2:** This is a very good question, and we suppose that different types of interleaved data may be suitable for different training tasks. Our results suggest that using filtered interleaved data like MMC4 may be better for vision model pre-training. We note that recent work OmniCorpus[a] shows that using original interleaved data like Obelics may be better for fine-tuning MLLM (see their Tab. 4).
>
> [a] OmniCorpus: An Unified Multimodal Corpus of 10 Billion-Level Images Interleaved with Text. arXiv:2406.08418.
> ___
> **Q3:** The scaling behavior is not demonstrated.
>
> **A3:** We are also very interested in the scaling behavior of our method. We anticipate using datasets such as Laion-2B and OmniCorpus[a] (a large-scale interleaved dataset) to train our vision model to achieve SoTA performance. However, using a total of 34B samples like OpenCLIP, training a ViT-L/14 model would require at least 8000 GPU days, which is more than we can afford now.
> ___
> **Q4:** Show the seen samples of each model.
>
> **A4:** The number of seen images is 13B for all models in Tab. 5 and 6, as mentioned in Section 4.3. For ablation studies, all models use 2B training images, as stated in Section 4.1.

---

> ### Comment · Reviewer_vQis · 2024-08-12
>
> Thank the authors for their response. And most of my concerns have been well-addressed. I'd like to maintain the rating in the current stage.

---

> > ### Author Response · Authors · 2024-08-14
> >
> > We are glad that your major concerns have been addressed. Thanks for your thorough and constructive review.

---

### Official Review · Reviewer_KZWZ · 2024-07-12

**Soundness:** 3
**Presentation:** 3
**Contribution:** 3
**Rating:** 4
**Confidence:** 4

**Summary:**

The paper tackles the problem of vision model pre-training. More exactly, it aims to exploit the interleaved image-text data that is very prevalent on the Internet. It proposes Latent Compression Learning that maximises the mutual information between the inputs and outputs of a causal attention model. When visual pre-training is applied to interleaved image-text data, visual latents are extracted using a visual encoding network and then combined with the text and fed to the causal model.

**Strengths:**

The paper tackles an important task and proposes an interesting method that may be of interest to the research community.

**Weaknesses:**

While the method seems interesting, my main concern is related to the experimental part that I find confusing. For example for BEiT3 the numbers reported are different from the ones reported in the paper.

Also, I think that for Tab 6, more multi-modal LLMs need to be included. While there can be a debate on fair vs unfair comparison, I think that you present results on a dataset these need to be complete. So, they can be greyed out, put in a different section, etc and explained why the comparison is not fair, but I don't think it's suitable for models that perform better to not be included at all. So, missing comparisons:

Fang, Yuxin, et al. "Eva: Exploring the limits of masked visual representation learning at scale." Proceedings of the IEEE/CVF Conference on Computer Vision and Pattern Recognition. 2023.
Zou, Xueyan, et al. "Generalized decoding for pixel, image, and language." Proceedings of the IEEE/CVF Conference on Computer Vision and Pattern Recognition. 2023.

or even some very recent ones for the sake of completeness:
Sun, Quan, et al. "Generative multimodal models are in-context learners." Proceedings of the IEEE/CVF Conference on Computer Vision and Pattern Recognition. 2024.
Liu, Haotian, et al. "Improved baselines with visual instruction tuning." Proceedings of the IEEE/CVF Conference on Computer Vision and Pattern Recognition. 2024.

**Questions:**

Why is OpenAI CLIP greyed out?

Why are the numbers reported for BEiT3 different than what they report in the paper? For example for Flicker 3K, R@1, it's reported 73.2 while the paper reports 88.2

Why, for example the comparison with BEIT3 is only shown in Tab.1 when they report results on VQAv2? The same question for CoCA?

**Limitations:**

Limitations are barely discussed at the end of the conclusions. Some limitations can be inferred from the rest of the paper.

---

> ### Author Rebuttal · Authors · 2024-08-07
>
> We thank the reviewer for the reviews and questions. But there is a misunderstanding here.
>
> First, as discussed in Fig. 1, we would like to clarify that our proposed LCL aims to pre-train a vision encoder from scratch using interleaved image-text data, rather than incrementally training a multi-modal model based on a pre-trained vision encoder. Their differences are illustrated in Fig. 1(b) and 1(c) and also acknowledged by Reviewer ktxP.
>
> Accordingly, our experiment is to evaluate the quality of pre-trained vision encoders rather than to compare the performances of different MLLMs. Therefore, when comparing with other pre-training methods, instead of using their released MLLMs checkpoints, we use their released codes or reproduce their methods to pre-train the visual encoders from scratch in a fair setting. We ensure a fair comparison in the same settings, including the model structure, training data, and evaluation settings.
> ___
> **Q1: (a)** For BEiT3, the numbers reported are different from the ones reported in the paper. **(b)** Why the comparison with BEIT3 is only shown in Tab.1 when they report results on VQAv2? The same for CoCA?
>
> **A1 (a):** The number of BEiT3 comes from training from scratch with BEiT3's pre-training task rather than the reported ones of the trained checkpoint. Tab. 1 compares pre-training methods and pre-training tasks, so the model and data for all methods should be fair. To avoid misunderstanding, we will clarify in the caption of Tab. 1 that the method names in parentheses refer to the pre-training tasks but not their trained checkpoints.
>
> **A1 (b):** Table 6 evaluates pre-trained vision encoders on multimodal dialogue tasks by integrating them with the same pre-trained LLM for fair comparison. Therefore, the reported results of BEiT3 and CoCa's released MLLM checkpoints are not included. In Table 1, BEiT3 and CoCa are compared as pre-training methods under fair settings.
> ___
> **Q2:** In Tab 6, more multi-modal LLMs need to be included.
>
> **A2:** As mentioned above, our proposed LCL pre-trains vision encoder from scratch, which is orthogonal to the training of MLLMs. Accordingly, our experiment is to evaluate the quality of pre-trained visual features rather than to compare the performances of different MLLMs.
>
> Tab. 6 is to fairly compare different pre-trained vision encoders in multimodal tasks, so the results of other MLLM's trained checkpoints, such as EVA[a], X-Decoder[b], and Emu[c], should not be incorporated. We have used LLaVA-1.5[d] for this evaluation (see Appendix A.2, "Multi-modal dialogue"). There are better MLLMs available than LLaVA-1.5, but training better MLLMs is beyond the scope of this paper, and we do not need to make such comparisons.
>
> In addition, we have compared the text generation + image feature regression task used by Emu in Tab. 1. However, EVA requires pre-trained CLIP features and X-Decoder uses segmentation data, so their training tasks are not comparable to those methods pre-training from scratch only using image-text data.
>
> [a] Eva: Exploring the limits of masked visual representation learning at scale. In CVPR, 2023
>
> [b] Generalized decoding for pixel, image, and language. In CVPR, 2023.
>
> [c] Generative multimodal models are in-context learners. In CVPR, 2024.
>
> [d] Improved baselines with visual instruction tuning. In CVPR, 2024.
> ___
> **Q3:** Why is OpenAI CLIP greyed out?
>
> **A3:** We aim to compare vision pre-training methods rather than pre-trained checkpoints. Both OpenCLIP and we use the LAION-400M dataset for training, allowing for a fair comparison. OpenAI CLIP, which uses private data, is included only as a reference.

---

> > ### Comment · Reviewer_KZWZ · 2024-08-10
> > **Rebuttal answer**
> >
> > Hi. Thank you for writing the rebuttal! I confirm I have read the rebuttal. I understand that the goal is to pre-train a vision encoder from scratch and the rebuttal sort of adds more clarity, but I think more details should be added in the paper to make things clearer. The authors have promised to add more details to the captions, so I don't have much to add here.
> >
> > However, when reporting results on a well defined benchmark, I personally believe all types of methods should be included, especially if they have a better performance. I understand that the space is limited and I do not propose to have a list of extensive non-related paper, but I consider including at least 1,2 especially can be very important for readers of the paper in order to make them aware of different methods and possibly increased performance on the benchmark. As I said in the original review, what is fair or not can be shortly explained and the methods clearly separated in the table.

---

> > > ### Author Response · Authors · 2024-08-14
> > >
> > > We sincerely thank you for your insightful comments and suggestions. Please see the general response. We revise our table to include more benchmark results with further discussion for a better understanding.

---

### Official Review · Reviewer_A7ig · 2024-07-13

**Soundness:** 3
**Presentation:** 2
**Contribution:** 2
**Rating:** 4
**Confidence:** 4

**Summary:**

The paper pre-trains models with a combination of a contrastive image-text objective and a generative language objective. The authors provide many results on image classification and vision-language tasks suggesting the competitiveness of the method in controlled settings.

**Strengths:**

S1. The paper is well framed and motivates nicely the need to pre-training on interleaved data.

S2. The paper gives good intuition about what the various equations mean, making the manuscript more accessible.

S3. Consideration of many pre-training datasets including LAION-400M, MMC4, and OBELICS.

S4. Extensive objective ablations spanning both contrastive and generative losses.

**Weaknesses:**

W1. [MAJOR] The paper presents the objective as novel (L44-54); however, it seems similar to CoCa (Yu et al., 2022.), which also employs a contrastive loss and a next token prediction loss. Can the authors clarify the differences and why the formulation is novel?

W2. It seems equation 3 appears in prior work; however, when it is first presented it seems to be presented as a novel insight. I recommend making the attribution to prior work more clear before introducing the equation.

W3. In the relation to previous pre-training tasks, it is important to also relate to CoCa. It seems the objective is pretty much the same suggesting that the objective is not actually a contribution of the work. Is there any reason CoCa is not mentioned here given the similarities?

W4. Make sure it is clear that you train on sequences with more than one image per sample (I am assuming this is true because you train on MMC4, but when explaining the objectives you include only one sequence for simplicity). 3.3 is a nice place to add this information. Also any special tricks to get multi-image to work? If so, it could also be nice to mention this.

W5. Why are the numbers for Flamingo in Tab 1 for IN-1k so low? Flamingo uses a pre-trained vision backbone, so I expect numbers to be good here.

W6. Is the COCO CIDEr evaluation protocol zero-shot? If so the number in table 4(a) of 87.5 looks extremely high relative to open flamingo and Idefics. Please double check this number and if few-shot prompting is used here, please make this clear. Also why is Gen. only worse than Con. only for captioning. How is contrastive learner able to do captioning?

W7. In the frozen transfer setting in Tab. 6 are all models fine-tuned on the same data? If so, what data? The specifics of the experiment are not clear to me, making it hard to interpret the results.

**Questions:**

Please see the weaknesses section for specific questions and topics to address.

**Limitations:**

The authors address limitations in a dedicated paragraph.

---

> ### Author Rebuttal · Authors · 2024-08-07
>
> We thank the reviewer for the careful reviews and constructive suggestions. We answer the questions as follows.
> ___
> **Q1: (a)** The proposed objective seems similar to CoCa, which also employs a contrastive loss and a next token prediction loss. Clarify the differences and why the formulation is novel. **(b)** In "relation to previous pre-training tasks", why CoCa is not mentioned given the similarities.
>
> **A1 (a):** CoCa cannot be extended to interleaved data since its learning objective relies on paired data. Applicable to interleaved data is a contribution of our proposed learning objective. There are significant differences between ours and CoCa in how the loss is applied:
> 1. CoCa's contrastive loss relies on the positive pairs from paired data, which cannot be directly obtained in general interleaved data.
> 2. CoCa's generation can only be conditioned on a single image, while our generation is conditioned on preceding context containing flexible interleaved image-text contents.
>
> **A1 (b):** In "relation to previous pre-training tasks," we aim to claim that existing pre-training methods cannot train visual features from scratch using interleaved data. CLIP is the most representative method for paired data, and CoCa is also specialized for paired data. Therefore, we did not specifically discuss CoCa in this context.
> Given the similarity in the loss form, we will include further discussions about CoCa in the revision and clarify our contribution as mentioned above.
> ___
> **Q2:** Equation 3 seems to appear in prior work. Make the attribution to prior work more clear before introducing the equation.
>
> **A2:** Thanks for your suggestion. Prior work (i.e., M3I mentioned in the paper) showed the relationship between cross-entropy and mutual information in other pre-training tasks, e.g., image self-supervised learning and contrastive learning. While Equation 3 specifically demonstrates this relationship when using an autoregressive model to compress interleaved image-text data. Our novel insight is that, in this scenario, the compression (cross-entropy) objective and the maximum mutual information objective are equivalent. In the revision, we will make the attribution to prior work and clarify our novel insight clearer.
> ___
> **Q3:** Make sure that it is trained on sequences with more than one image per sample. Any special tricks to get multi-image to work.
>
> **A3:** Yes, we support interleaved data containing multiple images. As described in Appendix A.1, each MMC4 sequence may contain 1 to 6 images. We did not use any special implementation tricks. See Fig. R2 in the rebuttal pdf for illustration. The visual features of multiple images are inserted at their positions in the sequence, combined with text embeddings, and then processed through a causal language model to compute the generation loss. For each image, its global feature and the language model’s output feature at its preceding <boi> token form a positive pair in contrastive loss. We will add this information to Section 3.3.
> ___
> **Q4:** Why are the numbers for Flamingo in Tab 1 for IN-1k so low?
>
> **A4:** The number of Flamingo comes from training from scratch with the Flamingo's pre-training task rather than its released checkpoint. Tab. 1 compares pre-training tasks, so the model and data for all methods should be fair. To avoid misunderstanding, we will clarify in the caption of Tab. 1 that the method names in parentheses refer to the pre-training task but not their trained checkpoints.
> ___
> **Q5: (a)** Is the COCO CIDEr evaluation protocol zero-shot? The number in Tab.4(a) of 87.5 looks extremely high. **(b)** Why is Gen. only worse than Con. only for captioning? How is contrastive learner able to do captioning?
>
> **A5 (a):** The results in Tab. 4 are evaluated under frozen transfer setting, which we will clarify in the table caption. See Appendix A.2  for frozen transfer details. Frozen transfer refers to fine-tuning downstream task models while the parameters of the pre-trained visual encoder are frozen. We use LAION-COCO for captioning fine-tuning, so the COCO CIDEr is reasonable.
>
> **A5 (b):** The vision encoder pre-trained by "Con. only"  also performs captioning by frozen transfer (see Appendix A.2). "Gen. only" is worse than "Con. only" for captioning, because visual features pre-trained by "Gen. only" experienced a collapse. We have discussed this feature collapse in Section 3.2, "Relation to Previous Pre-training Tasks."
> ___
> **Q6:** In the frozen transfer setting in Tab. 6, are all models fine-tuned on the same data? If so, what data?
>
> **A6:** Yes. The frozen transfer in Tab. 6 evaluates the pre-trained vision encoders on multimodal tasks following LLaVa-1.5. All models use the same data and training settings as LLaVa-1.5. See Appendix A.2 "Multi-modal dialogue" for more details.

---

> > ### Comment · Reviewer_A7ig · 2024-08-12
> >
> > Thanks for clarifying with respect to CoCa. I think given the similarities (e.g., Figure 1 of the manuscript and CoCa look very similar), it is important to make this comparison explicit. Also after reviewing other reviewers' comments, I am a bit concerned that state-of-the-art performance was not reported to contextualize the results. Additionally, many of the requested writing changes may require significant re-writing. Given this, I am electing to keep my score. Thanks to the authors for all of their effort in putting together the paper and the rebuttal.

---

> > > ### Author Response · Authors · 2024-08-14
> > >
> > > We sincerely thank you for your thorough and constructive review. In the revision, we will explicitly compare the differences between our objective and the CoCa's in the figure and text. Please see the general response for further discussion about the comparison with state-of-the-art performance.

---

### Official Review · Reviewer_ktxP · 2024-07-14

**Soundness:** 3
**Presentation:** 3
**Contribution:** 3
**Rating:** 5
**Confidence:** 4

**Summary:**

This paper aims to explore the use of weak supervision signals in multimodal interleaved image-text data to pretrain visual encoder, compressing the distribution of high-level features into the visual encoder. The paper employs contrastive loss and autoregressive loss to train the model. To prevent the collapse of visual representations, an entropy maximization constraint is applied. The paper derives the equivalence of maximizing the mutual information between the model's input and output as a latent compression and entropy constraint. The proposed pre-training method, called LCL, achieves performance comparable to CLIP on paired data while better utilizing the supervision information in interleaved data.

**Strengths:**

This paper explores how to use weak supervision signals in more general interleaved image-text data to accomplish visual pre-training. Its advantages are as follows:
1. Unlike previous approaches that fine-tune pre-trained visual models to align visual representations with the text space (Flamingo, LLaVA), this paper explores how to train visual models from scratch using interleaved image-text data. This is a meaningful exploration.
2. To prevent the collapse of visual representations, where autoregressive generation relies solely on textual information, this paper imposes an entropy constraint and further derives it as optimizing mutual information. This approach aids in model training.
3. Extensive quantitative experiments have validated the effectiveness of the visual models trained using this approach.

**Weaknesses:**

This paper has the following areas for improvement:
1. In some cases, the textual context may have little relevance to the image. It is worth investigating whether such data could harm the model's performance.
2. The paper lacks qualitative experiments to further demonstrate the effectiveness of the method. Designing reasonable visualization analyses would help to further elucidate the advantages of the approach.
3. Similar to CLIP, further demonstrating the model's transfer learning performance through domain adaptation tests and few-shot metrics would be beneficial.

**Questions:**

see weaknesses.

**Limitations:**

The paper's discussion on data bias and energy consumption limitations is commendable; however, it could further explore issues related to data privacy.

---

> ### Author Rebuttal · Authors · 2024-08-07
>
> Thanks for your good questions and constructive suggestions.
> ___
> **Q1:** In some cases, the textual context may have little relevance to the image. It is worth investigating whether such data could harm the model's performance.
>
> **A1:** This is a very good question.  It is inevitable that image-text data obtained from the internet may have weak correlations. For fair experiments, we currently follow the settings in previous works when using datasets.
> Previous works have used filtering methods to improve the quality of data. For example, the LAION-400M dataset used CLIP scores to filter image-text pairs, and OpenFlamingo used CLIP scores to filter out uncorrelated images in the MMC4 dataset. We now follow their settings.
>
> We note that recent work, OmniCorpus[a], analyzed model pre-training with differently filtered data in Section 5.2 (Table 3). The results show that appropriate filtering can improve the correlation of images and texts, thereby enhancing model performance. However, excessive filtering may lead to insufficient data diversity, thus hurting performance. We will analyze the impact of data quality on our model in future work.
>
> [a] OmniCorpus: An Unified Multimodal Corpus of 10 Billion-Level Images Interleaved with Text. arXiv:2406.08418.
> ___
> **Q2:** Qualitative experiments to further demonstrate the effectiveness of the method. Designing reasonable visualization analyses to further elucidate the advantages of the approach.
>
> **A2:** We use t-SNE to visualize the learned features using images from some classes in ImageNet-1K val set. See Fig. R1 in the rebuttal pdf. For both CLIP and our LCL, visual features of different classes are generally distinguishable, and the distance between classes is related to semantic similarity. Compared to CLIP, visualized points of LCL are slightly tighter for semantically similar classes and slightly further away for classes with low semantic relationships.
> ___
> **Q3:** Similar to CLIP, further demonstrating the model's transfer learning performance through domain adaptation tests and few-shot metrics.
>
> **A3:** Thanks for your suggestion. We adopt zero-shot domain adaptation tests in CLIP. Pre-training ViT-B/16 on LAION-400M, OpenCLIP achieves 67.1 on zero-shot ImageNet-1k, while ours is 60.1. However, domain adaptation via zero-shot retrieval is designed for contrastive learning such as CLIP. It may not be suitable for other types of pre-training methods (e.g., MIM, generative tasks). We have discussed this in Section 4.1.
>
> Moreover, similar to CLIP, we have used probing (fine-tuning a task head with backbone frozen) to evaluate transfer learning capabilities. See Tab.5, IN-1k frozen transfer results. Our LCL is comparable with OpenCLIP, demonstrating the strong transferability of our pre-trained vision model.

---

> > ### Comment · Reviewer_ktxP · 2024-08-12
> >
> > Thank you for your discussion on data quality and the analysis of LCL generalization performance. I believe that the contextual relevance of interleaved data (i.e., data quality) has a far greater impact on model performance than the quality of paired data. Furthermore, in real-world scenarios, the generalization performance of the model is a key metric for evaluating pre-trained representations, so it would be meaningful to analyze it.
> >
> > However, after reading the comments from other reviewers, I am concerned that the paper does not thoroughly discuss its relationship with other methods, especially regarding one of the core contributions of this paper—the pre-training objective. Additionally, as other reviewers have mentioned, further listing some state-of-the-art (SOTA) methods in the evaluation and clarifying how to achieve a fair comparison would further reduce misunderstandings for readers. Considering the above factors, I will somewhat lower my rating. Further discussion has the potential to affect my ratings.

---

> > > ### Author Response · Authors · 2024-08-14
> > >
> > > We sincerely thank you for your thorough review and constructive comments. We respond to your concerns in the general response.

---

### Author Rebuttal · Authors · 2024-08-07

We thank all the reviewers for the careful reviews and constructive suggestions. We respond to your questions respectively. The PDF contains supplementary figures for rebuttal.

---

> ### Author Response · Authors · 2024-08-14
> **General Response**
>
> We respond to the common concerns of the reviewers here.
>
> First, we claim that our proposed method is the first to use interleaved image-text data to pre-train vision encoder from scratch. Existing vision pre-training methods do not support interleaved data. Simple ways to use interleaved data, e.g., autoregressive text generation, do not perform well for vision pre-training due to the vision feature collapse. We address this by our novel training objective that leverages both contrastive and generation loss.
> ___
> **Q1:** Comparison with CoCa's objective.
>
> **A1:** The major difference between our approach and CoCa's objective is that CoCa does not support interleaved data. For a detailed discussion, please refer to the response to Reviewer A7ig. In the revision, we will further elaborate on the differences between our method and CoCa, both in the figures and the textual descriptions.
> ___
> **Q2:** Comparison with state-of-the-art methods.
>
> **A2:** We would like to emphasize that we propose a vision pre-training method with image-text data. Accordingly, it is appropriate to benchmark it against SoTA vision pre-training methods that use image-text data. We suppose that achieving SoTA performance at the system level of MLLM is not a requisite for a vision pre-training method.
> Below are two tables revised from the originally tables in our paper, which we will use to further explain our comparison with SoTA performance.
>
> 1. Existing vision pre-training methods only support paired data, and we compare with their SoTA performance on paired data. Due to the unavailability of some data, checkpoints, and source code, we have had to either reproduce existing methods or rely on available open-source implementations for a fair comparison (marked by * in Tab R1). The results in Tab. R1 show that our method is comparable to existing methods on paired data.
>
> 2. To show that using interleaved image-text data for vision pre-training is not trivial, we also consider existing ways to use interleaved data. Those methods are proposed as MLLM training methods, but we test them for vision pre-training. Tab R1 shows those methods do not work well for vision pre-training.
>
> 3. To further confirm our proposed method, we need to compare our pre-trained model with existing checkpoints. Tab. R1 shows that our pre-trained ViT-L/14 is on par or better than the OpenCLIP's checkpoint in a fair comparison. Other checkpoints listed in Tab. R1 could be the reference. They either use a larger vision encoder, more training data, or private data.
>
> 4. For multimodal benchmarks, as explained in the response to reviewer KZWZ, our goal is to fairly evaluate pre-trained vision encoders, not to compare different MLLMs at the system level. It is reasonable for us to adopt the open-source LLaVA-1.5. We also list some advanced MLLMs for reference in Tab. R2. Those MLLMs may outperform ours due to their use of stronger vision encoders, larger LLMs, or private data. We anticipate constructing a SoTA MLLM system in future work, but in the current stage, we focus on vision model pre-training due to resource constraints.

---

> > ### Author Response · Authors · 2024-08-14
> > **General Response cont.**
> >
> > | Vision pre-training method | Support interleave data | Vision encoder | Pre-training data | Seen samples | Data public | IN-1k frozen | IN-1k finetune | COCO ret. frozen TR@1 | COCO ret. frozen IR@1 | COCO cap. finetune CIDEr |
> > |---|---|---|---|---|---|---|---|---|---|---|
> > | Existing vision pre-training methods only support paired data |  |  |  |  |  |  |  |  |  |  |
> > | LCL (ours) | ✅ | ViT-B (86M) | LAION-400M | 2B | ✅ | 75.0 |  | 48.3 | 34.3 |  |
> > | *OpenCLIP | ❌ | ViT-B (86M) | LAION-400M | 2B | ✅ | 75.0 |  | 47.2 | 34.2 |  |
> > | *Reproduced CoCa | ❌ | ViT-B (86M) | LAION-400M | 2B | ✅ | 75.2 |  | 48.6 | 34.8 |  |
> > | Some MLLM training methods support interleaved data but do not work for vision pre-training |  |  |  |  |  |  |  |  |  |  |
> > | LCL (ours) | ✅ | ViT-B (86M) | MMC4 | 2B | ✅ | 75.2 |  | 48.5 | 34.5 |  |
> > | *BEiT3 | ✅ | ViT-B (86M) | MMC4 | 2B | ✅ | 73.3 |  | 45.1 | 30.6 |  |
> > | *OpenFlamingo | ✅ | ViT-B (86M) | MMC4 | 2B | ✅ | 24.0 |  | 10.6 | 5.6 |  |
> > | *Emu | ✅ | ViT-B (86M) | MMC4 | 2B | ✅ | 5.7 |  | 2.3 | 1.4 |  |
> > | Compare with existing checkpoints or reported results |  |  |  |  |  |  |  |  |  |  |
> > | LCL (ours) | ✅ | ViT-L (304M) | LAION-400M | 13B | ✅ | 82.2 | 86.1 | 59.6 | 46.2 | 129.7 |
> > | LCL (ours) | ✅ | ViT-L (304M) | LAION-400M + MMC4 | 13B | ✅ | 82.0 | 86.1 | 60.0 | 46.0 | 130.9 |
> > | OpenCLIP | ❌ | ViT-L (304M) | LAION-400M | 13B | ✅ | 82.1 | 86.2 | 59.5 | 46.0 | 128.9 |
> > | OpenAI CLIP | ❌ | ViT-L (304M) | WIT-400M | 13B | ❌ | 83.7 | 87.4 | 61.7 | 48.2 | 132.7 |
> > | EVA-02-CLIP | ❌ | ViT-L (304M) | $\dagger$ WIT-400M, LAION2B, etc | ~23B | ❌ |  | 90.0 |  |  |  |
> > | CoCa | ❌ | ViT-G (1B) | JFT-3B + ALIGN | 33B | ❌ | 90.6 | 91.0 |  |  | 143.6 |
> >
> > Tab. R1: Comparison of different vision pre-training methods and models. Revised from Tab. 1, 2, 5 in the paper. * means reproducing or using open source code. $\dagger$ Training data of the teacher model is also considered.

---

> > > ### Author Response · Authors · 2024-08-14
> > > **General Response cont. 2**
> > >
> > > | Vision pre-training method | Vision encoder | Pre-training samples | Pre-training data public | MLLM method | LLM size | MLLM training samples | MLLM data public | VQAv2 | GQA | VizWiz | SQA | POPE | MME | MMB | SEED image |
> > > |---|---|---|---|---|---|---|---|---|---|---|---|---|---|---|---|
> > > | Our transfer evaluation |  |  |  |  |  |  |  |  |  |  |  |  |  |  |  |
> > > | LCL (ours) | ViT-L (304M) | 13B | ✅ | LLaVA-1.5 | 7B | 1.2M | ✅ | 70.7 | 57.4 | 41.4 | 69.7 | 81.7 | 1291.7 | 55.3 | 56.5 |
> > > | OpenCLIP | ViT-L (304M) | 13B | ✅ | LLaVA-1.5 | 7B | 1.2M | ✅ | 68.7 | 57.0 | 39.5 | 69.0 | 81.8 | 1266.2 | 54.2 | 55.8 |
> > > | OpenAI CLIP | ViT-L (304M) | 13B | ❌ | LLaVA-1.5 | 7B | 1.2M | ✅ | 77.1 | 61.7 | 44.4 | 71.1 | 84.6 | 1486.9 | 65.1 | 64.6 |
> > > | Advanced MLLMs as reference |  |  |  |  |  |  |  |  |  |  |  |  |  |  |  |
> > > | OpenAI CLIP | ViT-L-336 (304M) | 13B | ❌ | LLaVA-1.5 | 7B | 1.2M | ✅ | 78.5 | 62.0 | 50.0 | 66.8 | 85.9 | 1510.7 | 64.3 | 65.4 |
> > > | OpenAI CLIP | ViT-L-336 (304M) | 13B | ❌ | LLaVA-NEXT | 7B | 1.3M | ❌ | 81.8 | 64.2 | 57.6 | 70.1 | 86.5 | 1519 | 67.4 | 70.2 |
> > > | OpenAI CLIP | ViT-L (304M) | 13B | ❌ | InternLM-XC2 | 7B | ~20M | ❌ |  |  |  |  |  | 1712 | 79.6 | 75.9 |
> > > | OpenCLIP | ViT-L (304M) | 34B | ✅ | Qwen-VL-Chat | 7B | ~1.5B | ❌ | 78.2 | 57.5 | 38.9 |  |  | 1487.5 |  |  |
> > > | EVA-CLIP | ViT-G (1B) | ～28B | ❌ | *Emu-I | 13B | ~83M | ❌ | 62.0 | 46.0 | 38.3 |  |  |  |  |  |
> > > | EVA-CLIP | ViT-E (4.4B) | ～28B | ❌ | *Emu2-Chat | 33B | ~160M | ❌ | 84.9 | 65.1 | 54.9 |  |  |  |  |  |
> > > | EVA-CLIP | ViT-E (4.4B) | ～28B | ❌ | CogVLM-Chat | 7B | ~1.5B | ❌ | 82.3 |  |  | 91.2 | 87.9 |  | 77.6 |  |
> > > | InternViT | ViT-6B | ～29B | ❌ | InternVL | 7B | ~1.0B | ❌ | 79.3 | 62.9 | 52.5 |  | 86.4 | 1525.1 |  |  |
> > > | InternViT | ViT-6B | ～29B | ❌ | InternVL 1.5 | 20B | ~25M | ❌ | - | 65.7 | 63.5 | 94.0 | 88.3 | 1637 | 82.2 | 76.0 |
> > >
> > > Tab. R2: Results of multimodal tasks evaluation. Revised from Tab. 6 in the paper.

---

### Decision · Program_Chairs · 2024-09-25

**Decision:**

Accept (poster)

**Comment:**

The paper received originally mixed comments. Although all reviewers agreed that the direction is useful, and the approach is technically sound, there are two main issues raised: the similarty of the training objective proposed to CoCa, and the apparent discrepancy of reported SoA results to published ones. Both issues were discussed in length between the authors and reviewers, and the reviewers offered convincing explanations and new information for both.

I lean towards accepting this paper, assuming that the new information and clarifications explosed during the review process will be properly reflected in the revised version.